

# Harmful effects of the microplastic pollution on animal health: a literature review

Natalia Zolotova[1], Anna Kosyreva[1,2], Dzhuliia Dzhalilova[1], Nikolai Fokichev[3] and Olga Makarova[1]

[1] Avtsyn Research Institute of Human Morphology of Federal State Budgetary Scientific Institution, "Petrovsky National Research Centre of Surgery", Moscow, Russia
[2] Medical Institute, RUDN University, Moscow, Russia
[3] Biological Department, Lomonosov Moscow State University, Moscow, Russia

## ABSTRACT

**Background:** The environmental pollution by microplastics is a global problem arising from the extensive production and use of plastics. Small particles of different plastics, measured less than 5 mm in diameter, are found in water, air, soil, and various living organisms around the globe. Humans constantly inhale and ingest these particles. The associated health risks raise major concerns and require dedicated evaluation.

**Objectives:** In this review we systematize and summarize the effects of microplastics on the health of different animals. The article would be of interest to ecologists, experimental biologists, environmental physicians, and all those concerned with anthropogenic environmental changes.

**Methodology:** We searched PubMed and Scopus from the period of 01/2010 to 09/2021 for peer-reviewed scientific publications focused on (1) environmental pollution with microplastics; (2) uptake of microplastics by humans; and (3) the impact of microplastics on animal health.

**Results:** The number of published studies considering the effects of microplastic particles on aquatic organisms is considerable. In aquatic invertebrates, microplastics cause a decline in feeding behavior and fertility, slow down larval growth and development, increase oxygen consumption, and stimulate the production of reactive oxygen species. In fish, the microplastics may cause structural damage to the intestine, liver, gills, and brain, while affecting metabolic balance, behavior, and fertility; the degree of these harmful effects depends on the particle sizes and doses, as well as the exposure parameters. The corresponding data for terrestrial mammals are less abundant: only 30 papers found in PubMed and Scopus deal with the effects of microplastics in laboratory mice and rats; remarkably, about half of these papers were published in 2021, indicating the growing interest of the scientific community in this issue. The studies demonstrate that in mice and rats microplastics may also cause biochemical and structural damage with noticeable dysfunctions of the intestine, liver, and excretory and reproductive systems.

**Conclusions:** Microplastics pollute the seas and negatively affect the health of aquatic organisms. The data obtained in laboratory mice and rats suggest a profound negative influence of microplastics on human health. However, given significant

Corresponding author
Anna Kosyreva, kosyreva.a@list.ru

variation in plastic types, particle sizes, doses, models, and modes of administration, the available experimental data are still fragmentary and controversial.

# INTRODUCTION

The term "plastics" refers to artificial materials based on synthetic or natural high molecular weight compounds—polymers. Starting from the 1950s, plastics have been extensively used in many industries due to their economic effectiveness, versatility, lightness, strength, and durability. Of the more than 6.3 billion tons of plastics produced globally over the past 60 years, about 9% were recycled into secondary raw materials and about 12% were utilized by incineration (*Alabi et al., 2019*). The plastic waste degrades very slowly; decomposition of a 1 mm thick plate takes from several decades to a hundred years, depending on its chemical content and environmental conditions (*Chamas et al., 2020*). During decomposition, plastic products disintegrate into small pieces. Plastic fragments less than 5 mm in size are termed "microplastics" (MP). Several studies suggest a separate term "nanoplastics" (NP) for the plastic particles measured less than 100 nm in diameter. By origin, MP are subdivided into: (1) primary MP—deliberately fabricated microparticles for consumer and industrial purposes, to be used as cleaning abrasives, cosmetics, polymer carriers for drug delivery, sandblasting agents, plastic-coated fertilizers, *etc.*; and (2) secondary MP produced spontaneously upon disintegration of the bulk plastic waste (*Hirt & Body-Malapel, 2020*; *Yee et al., 2021*).

Due to the small size of the particles, MP are rapidly spread with wind and water. As a result, the particles are found in air, soil, water, and polar ice, in the depths of the sea, and in living organisms (*Fackelmann & Sommer, 2019*).

The effects of MP on living organisms depend on the physical and chemical natures of the particles. Chemical properties of MP reflect the chemical nature of the polymer and properties of the additives. The polluting MP are predominated by polyethylene (PE), polypropylene (PP) and polystyrene (PS) particles. The additives include inert or reinforcing fillers which endow the plastic with specific properties: plasticizers, antioxidants, UV stabilizers, lubricants, dyes, and flame retardants. Physical properties of MP include size and shape, the elasticity, shear strength, and surface charge of the particles. By shape, MP are subdivided into fibers, grains, granules, fragments, films, and foams. The high surface-to-volume ratio of MP makes them capacious adsorbents of pollutants and microorganisms (*Campanale et al., 2020*; *Wright et al., 2020*).

In this review, we aimed to systematize and summarize the experimental studies evaluating the health-related effects of MP in different animals in order to estimate the corresponding risks to human health. We discuss the levels of environmental pollution with MP, estimate the consumption of MP by humans, and describe the effects of MP on animal health.

A special emphasis is made on experimental studies of MP effects in laboratory rodents. Mice and rats undoubtedly remain the main model organisms in the research on human diseases. Rodent models have been widely applied to understand the mechanisms of pathogenesis, to test the efficacy of candidate drugs, and to predict the side effects and individual responses. The harmful MP effects in mice and rats are closely related to the corresponding risks in humans.

## SURVEY METHODOLOGY

We searched PubMed and Scopus databases for the peer-reviewed articles focused on (1) environmental pollution with MP; (2) MP uptake by humans; and (3) the impact of MP on animal health, published from 01/2010 to 09/2021.

1. We used "microplastic pollution" as a basic query and added terms "soil", "air", "water", or "ocean" for detalization.
2. We used "microplastic food", "microplastic human", and "microplastic human consumption" queries to search for information about the consumption of MP by humans.

The search returned a huge variety of original research articles and reviews. Of the 3,569 publications retrieved from PubMed for "microplastic pollution" with an applied filter "in the last 5 years" <2017–2021>, 584 articles were classified as reviews, systematic reviews, or meta-analyses. The "microplastic food" query returned 986 publications, 194 of them reviews; the "microplastic human" query—1,040 and 292 items, respectively. In writing this article, we focused on the global character of the environmental pollution with MP and its impact on human health in order to highlight the relevance of the problem. Accordingly, we tended to include the most recent items. A total of 20 review articles describing the spread of MP in the environment, living organisms, and human foods were surveyed for this part of the study.

3. We used "microplastic animal", "microplastic hazard", or "microplastic toxicity" primary queries to figure out information about the impact of MP on animal health. We also limited the search results to the most recent items.

According to the search results, the focus on potential hazards of MP to aquatic organisms has quite a history. A PubMed search over "the last 5 years" (2017–2021) retrieved over 700 publications concerning the effects of MP specifically in fish, 73 of them classified as reviews. In this study we provide a brief survey of recent reviews on the harmful effects of MP in aquatic invertebrates (*Haegerbaeumer et al., 2019*) and fish (*Yong, Valiyaveetill & Tang, 2020*).

Considering the common involvement of mice and rats as *in vivo* toxicity models, we eventually narrowed the focus to the toxic effects of MP in laboratory rodents. Accordingly, PubMed and Scopus databases were searched with "microplastic mouse" and "microplastic rat" queries over a period from 01/2010 to 09/2021.

The inclusion criteria were as follows: experimental research involving mice and rats, published in English and listed in MEDLINE (PubMed) or Scopus starting from 01 January 2010; the original experimental *in vivo* studies featuring polymer microparticles of any type.

The exclusion criteria were as follows: publications prior to January 1st, 2010; unavailability of English version of the text; and/or *ex vivo*/*in vitro* studies. We also excluded commentaries, summaries, reviews, editorials, and duplicate studies.

The earliest article that complied with these criteria was published in 2017. A total of 30 articles published in 2017–2021 matched the specified requirements. We classified and summarized the parameters of MP exposure (plastic type, particle size, administration route, dose, and duration) and its effects on animal health.

## RESULTS

### MP content measurement

Detection of MP in various natural and biological samples is not a straightforward task. Non-homogeneous distributions of the particles, variability of their chemical composition, shape and size, as well as the presence of naturally occurring fractions that are difficult to distinguish from the synthetic plastic particles, significantly complicate the evaluation of MP content in various media. Of note, MP content of similar samples obtained from different sources may vary significantly. For example, MP content of kitchen salt, depending on its production source and the method used for MP detection, ranges from zero (absence of detectable MP) to 5,400 particles per kilogram (*Kwon et al., 2020*).

The correct MP content evaluation implies elimination of natural organic polymers prior to the measurements. For liquid samples such as water, dissolved sea salt, and honey, a pretreatment with hydrogen peroxide is usually sufficient. Alkalis (KOH, NaOH), Fenton's reagent ($Fe^{2+}$ ion with $H_2O_2$), acids ($HNO_3$, $HClO_4$), and digestive enzymes (proteinase, trypsin, and collagenase) are applied to isolate MP from living tissues on top of the $H_2O_2$ pretreatment. The pretreated samples are passed through a filter. The MP particles precipitated on the filter are counted using a microscope. The type of plastic is identified by Fourier Transform Infrared Spectroscopy or Raman spectroscopy (*Kwon et al., 2020*).

### Environmental pollution with MP

Approximately 60–80% of the garbage collected on the earth consists of plastic and almost 10% of the global production of plastics end up in the oceans where their decomposition may take several hundred years (*Avio, Gorbi & Regoli, 2017*). An estimated >8 million tons of plastic enter the oceans annually (*Erni-Cassola et al., 2019*). Quantification of the global burden of oceanic plastic is difficult and controversial. According to rough estimates, the world's oceans contain at least 5.25 trillion plastic particles weighing 269 thousand tons in total (*Eriksen et al., 2014*). Remarkably high concentrations of plastic waste are observed in the central North Pacific Ocean. This infamous accumulation of anthropogenic debris is known as the Great Pacific Garbage Patch, or the Eastern Garbage Continent. The patch of about 1.6 million $km^2$ in size contains at least 79 thousand tons of

plastic, about 8% of which is constituted by MP (*Lebreton et al., 2018*). A total of five oceanic circulations with plastics have been identified to-date (located in the North and South Atlantic Ocean, South Indian Ocean, and North and South Pacific Ocean) (*Avio, Gorbi & Regoli, 2017*). The most common types of plastics encountered in the marine environments are polyethylene (PE, about 23% of plastic particles); polyesters-polyamides-acrylics (PP & A, 20%); polypropylene (PP, 13%); and polystyrene (PS, 4%). Concentrations of plastic particles are about $10^3$–$10^4$ particles/m$^3$ in tidal sediments, 0.1–1 particles/m$^3$ in surface waters (predominantly the low-density PE and PP), and over $10^4$ particles/m$^3$ in deep-sea sediments (PE- and PP-free, dominated by PP & A and chlorinated polyethylene, CPE) (*Erni-Cassola et al., 2019*).

Rivers are the main routes of transportation of the plastic waste from land to the sea, bringing 1.15–2.41 million tons of plastic waste to the world's ocean a year. MP content in fresh water reservoirs of North and South America varies from 0.16 to 3,438 particles/m$^3$, in European rivers—from 0.28 to 1,265 particles/m$^3$, and in Asian water bodies—from 293 to 19,860 particles/m$^3$ (*Sarijan et al., 2021*). In these aquatics, MP are represented mainly by fibers, shreds, films, and foams composed of PP, PE, and PS (*Sarijan et al., 2021*). Despite all efforts applied to remove MP from tap water, the particles are definitely there. An analysis of 159 samples of tap water collected at different regions of the world detected MP in 81% of the samples; the average content was 5.45 particles/L (*Kosuth, Mason & Wattenberg, 2018*).

MP are ubiquitously found in soil environments including farmlands, greenhouses, home gardens, as well as coastal, industrial, and floodplain soils (*Hirt & Body-Malapel, 2020*). In agricultural soils, the main sources of MP pollution are solid biological substances and composts, irrigating sewage waters, mulch films, polymer fertilizers and pesticides, and rainfall (*Kumar et al., 2020*). In the floodplain soils of Sweden, MP content reaches 55.5 mg/kg, or 593 particles/kg (*Scheurer & Bigalke, 2018*). The plastics in soils are mainly represented by PE, PP, PS, and polyvinyl chloride (PVC) (*Kumar et al., 2020*).

Needless to say, MP particles are present in the air. The average rates of MP deposition in urban and suburban areas in Paris constitute 110 ± 96 and 53 ± 38 particles per m$^2$/day, respectively (*Dris et al., 2016*), whereas for Dongguan (China) and central London the rates are 36 ± 7 particles per m$^2$/day (*Cai et al., 2017*) and 771 ± 167 particles per m$^2$/day, respectively (*Wright et al., 2020*). Atmospheric MP are mainly acrylic fibers, 5–75 μm thick and 250–2,500 μm long, as well as non-fibrillar shreds of PE, PP, and PS measured 50–350 μm (*Wright et al., 2020*). The atmospheric MP fibers come from textiles and the non-fibrous particles mostly come from decomposing packaging materials; other prominent sources include worn-out tires, paints, and industrial emissions (*Wright et al., 2020*).

## MP in the food chains and human foods

Under the progressive intensive pollution of all habitats by MP, the particles inevitably end up in animal bodies. MP uptake can be direct (from media) or indirect (from prey, *i.e.*, transmitted along the food chains) (*Smith et al., 2018*). MP are most readily accumulated in the bodies of biofiltering animals—mollusks, ascidia, and zooplankton. MP content

in soft tissues of blue mussels and giant Pacific oysters is 36 ± 7 and 47 ± 16 particles/100 g, respectively, whereas an average fish carries 35 MP particles in the gastrointestinal tract (*Van Cauwenberghe & Janssen, 2014*; *Kwon et al., 2020*).

MP also pervade in terrestrial food chains; they were detected in chicken stomachs (5.1 particles/g) and feces (105 particles/g) (*Huerta Lwanga et al., 2017*), as well as in sheep feces (1,000 particles/g) (*Beriot et al., 2021*).

Human beings ingest MP with food and water, and also inhale them with the air. MP particles are found in sugar (0.44 particles/g), honey (0.10 particles/g), salt (0.11 particles/g), alcohol (32.27 particles/L), bottled water (94.37 particles/L), tap water (4.23 particles/L), and inhaled air (9.80 particles/$m^3$) (*Cox et al., 2019*; *Yee et al., 2021*). *Danopoulos et al. (2020)* reported high content of MP in seafood: an individual consumes up to 27,825 particles of MP with shellfish, up to 17,716 particles with crustaceans, and up to 8,323 particles with fish a year; the total individual consumption of MP with seafood may reach 53,864 particles a year.

According to *Kwon et al. (2020)*, over 50% of the MP in food are composed of PE, PP, PS, and polyethylene terephthalate (PET). These particles are predominantly fibers including thin filaments. The fibers are considered to cause toxic effects at lower doses than spherical particles (*Kwon et al., 2020*).

As estimated by *Cox et al. (2019)*, an average individual consumes from 39,000 to 52,000 MP particles a year with food and drink. If we add the inhalable airborne plastic particles, the total number of plastic particles entering the human body will reach 74,000–121,000 particles a year (*Yee et al., 2021*). However, these estimates are indirect and too approximate, requiring further specification.

MP particles were found in human feces: an average of twenty 50–500 μm particles per 10 g, chiefly PP and PET (*Schwabl et al., 2019*), and single MP particles were found in human placenta (*Ragusa et al., 2021*).

## MP effects in animals
### Invertebrates

The effects of MP exposure have been characterized in detail for a number of aquatic species. *Haegerbaeumer et al. (2019)* meticulously reviewed the studies on the impact of MP on the benthic marine and freshwater invertebrates including annelids, arthropods, ascidians, sea urchins, bivalve mollusks, and rotifers. Of the 28 studies assessing the effect of MP on mortality, only three revealed significant changes: the mortality increased in polychaetas *Perinereis aibuhitensis* (PS ø 8–12 μm, 100–1,000 particles/mL), shrimps *Palaemonetes pugio* (PE, PS, PP ø 30–165 μm, 50,000 particles/L), and copepods *Tigriotopus japonicus* (PS ø 0.05 μm, 1.25 μg/L). MP exposure promoted a decline in food activity and fertility; it also inhibited the larval growth and development and increased oxygen consumption and reactive form production. However, given significant variation of types, sizes, and concentrations of MP particles among the studies, as well as the use of different animal phyla, a dedicated comparative analysis of MP effects based on these studies will hardly make sense.

### Fish

MP were found in 49% of the examined fish from the North East Atlantic Ocean (*Dicentrachus labrax*, *Trachurus trachurus*, and *Scomber colias*). Approximately 35% of the examined specimens had MP in the gastrointestinal tract (mean ± SD = 1.2 ± 2.0 items/individual), 36% had MP in the gills (0.7 ± 1.2 items/individual), and 32% had MP in the dorsal muscles (0.054 ± 0.099 items/g of tissue). The MP particles were mainly represented by 151–1,500 μm fibers and 100–1,500 μm shreds of PE and polyester. In fish, MP exposure leads to increased lipid peroxidation levels in the brain, gills and spinal muscles; it also stimulates the activity of brain acetylcholinesterase (*Barboza et al., 2019*).

*Yong, Valiyaveetill & Tang (2020)* conducted a meta-analysis of the experimental studies revealing significant toxic or pathological MP effects in fish. Toxic reactions generally arise from smaller particles (≤5 μm in diameter), whereas bigger PS particles (100 μm or more) have no significant effect. During the tests, MP particles are usually added to a water tank containing fish and roe at concentrations varying from 1 to 1,000 mg/L, most conventionally at 20 mg/L. The time of exposure varies from several hours to several months; the most conventionally used exposure time is 1 week. MP uptake has been shown to disturb the feeding behavior and motor activity in adult fish and fry, and also to negatively affect fertility. Maternal transmission of MP to offspring has been documented (*Pitt et al., 2018a*), along with an evidence that prenatal exposure of MP affects the early fry development (*Wang et al., 2019*). Increasing doses of MP exacerbate the degree of their accumulation in tissues and severity of the associated histological and biochemical changes (*Lu et al., 2016*; *Wang et al., 2019*; *Pannetier et al., 2020*).

The majority of experimental studies on MP in fish have been performed on zebrafish using PS or, less commonly, PE particles. The 25–70 nm PS particles penetrate the roe membrane, accumulate in the yolk sac, and move to the gastrointestinal tract. In fish embryos and fry, the nanoparticles are found in the intestines, gallbladder, liver, pancreas, heart, and brain. The transfer of PS nanoparticles from mother to the offspring has been confirmed experimentally. The nanoparticles have been shown to interfere with fry motility, causing either hyperactivity or weakness, boost cortisol levels, suppress glucose levels, and increase the rates of heartbeat, inflammatory reactions, and fatty degeneration of the liver. The nanoparticles also affect the antioxidant system by inhibiting glutathione reductase in the brain, muscles, and testes of the adult fish and their offspring; they interfere with synaptic transmission by suppressing acetylcholinesterase activity and inhibit the expression of cytoskeleton genes (*Lu et al., 2016*; *Chen et al., 2017*; *Pitt et al., 2018a*; *Pitt et al., 2018b*; *Brun et al., 2019*).

Larger PS particles, 5 μm in diameter, found in the gills, intestines, and livers of fry and adult fish, promote fatty degeneration of hepatocytes and inflammatory changes in the liver and intestines, alter the qualitative and quantitative composition of the intestinal microbiome, interfere with carbohydrate and lipid metabolism, and induce the oxidative stress-related changes in the antioxidant protection gene expression (*Lu et al., 2016*; *Qiao et al., 2019*; *Wan et al., 2019*).

Yet bigger particles of PS and PE, 10–90 μm in diameter, accumulate in the intestines of fry and adult fish and in the gills of the adults. These particles cause changes in fish behavior and intestinal microbiome composition; they enhance the rates of neutrophil infiltration in the intestinal mucosa and the gills, interfere with lipid, carbohydrate, amino acid, and nucleic acid metabolism, affect the antioxidant defense mechanism, reduce the post-hatching survival, and inhibit regulatory genes involved in development and functioning of the nervous system (*Lu et al., 2016*; *LeMoine et al., 2018*; *Limonta et al., 2019*; *Mak, Yeung & Chan, 2019*; *Malafaia et al., 2020*; *Wan et al., 2019*).

The second most popular experimental models among fishes are rice fish, or medaka (*Oryzias latipes* and *Oryzias melastigma*). After exposure to PS particles (0.05 or 10 μm) or "natural" MP samples collected on the beach, the specimens presented with MP accumulations in the gills, intestines, and liver. The exposure led to increased mortality, a decrease in the average length and weight of fry and adult fish, oxidative stress and structural tissue damage at the sites of MP accumulation, increased content of reactive oxygen species and altered antioxidant enzyme activities, reproductive dysfunctions, and a decrease in roe production. Prenatal exposure to MP affected the early offspring development (*Cong et al., 2019*; *Wang et al., 2019*; *Zhu et al., 2019*; *Pannetier et al., 2020*).

Similar experiments involving other fish species (crucian carp, common carp, tilapia, catfish, perch, dorada, minnow) revealed the accumulation of MP (PS 0.025–15 μm; PE 40–100 μm, or PVC 40–200 μm) in the intestines, gills, liver, and brain. The exposure caused changes in feeding and shallow water behaviors, a decrease in swimming speed and range of movement, and fry growth retardation. At the tissue level, the exposure led to pronounced metabolic imbalance, with increased oxygen consumption and ammonia excretion, changes in the activity of liver enzymes, oxidative stress, structural damage to the liver, gills, intestines, skin and muscles, suppression of acetylcholinesterase activity in the brain, and altered expression of genes of the reproductive axis (*Yong, Valiyaveetill & Tang, 2020*).

It should be stressed that, in fish, MP accumulate mainly in the intestine, and in certain cases also in the gills and the liver, causing primary pathological changes in these organs. The compromised integrity of the epithelial barrier in the gut affects gene expression and protein production profiles, boosts the inflammation and oxidative stress levels, and alters gut microbiota. The signs of imbalanced lipid and carbohydrate metabolism and oxidative stress have been also observed in the affected fish livers (Fig. 1).

In addition, MP may enhance the toxic effects of certain pollutants/toxic substances in fish, including phenanthrene (*Karami et al., 2016*), mercury (*Barboza et al., 2018*), cadmium (*Lu et al., 2018a*; *Banaee et al., 2019*; *Miranda, Vieira & Guilhermino, 2019*), polychlorinated biphenyls (*Rainieri et al., 2018*), gold ions (*Lee et al., 2019*), and antibiotics (*Zhang et al., 2019*; *Yong, Valiyaveetill & Tang, 2020*).

### Mammals

Studies focused on MP effects in terrestrial mammals were represented by 30 articles on the effect of MP particles in laboratory mice and rats listed in PubMed and Scopus databases. The earliest was published in 2017 and about half of the rest were published in

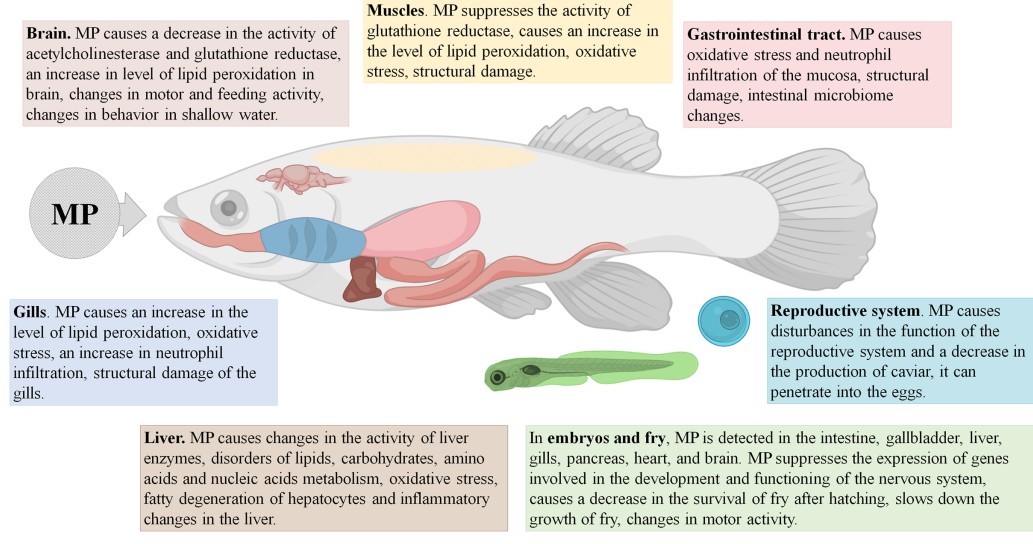

**Figure 1 Effects of microplastics (MP) on fishes.**

2021, which indicates the growing interest of the scientific community towards this problem.

It should be noted that species and strains of the animals, as well as doses, types, exposure lengths, and administration routes of the particles, varied considerably among the studies (Table 1). The majority of studies retrieved by the search (24 articles) involved mice, mostly ICR (CD-1) or C57Bl/6 and less often Balb/c or Swiss strains, whereas six of the studies were performed on Wistar and Sprague-Dawley rats. The administration routes were mostly oral, with the water from a drinking bowl (13 studies) or through a gastric tube (13 studies). In the rest few studies, MP were administered with food, by inhalation, intratracheally, or intraperitoneally. The particles were predominantly PS with diameters ranging from 0.02 to 500 µm (4.5 µm on average) (25 studies) and less often PE measured 0.4–150 µm (35 µm on average) (six studies).

The doses for oral exposure varied from 0.01 to 100 mg MP per kg of body weight a day. The daily doses of MP most typically used in mice were 0.024 or 0.24 mg/kg when added to a drinking bowl and 4 or 20–60 mg/kg when administered through a gastric tube; in rats, the typical doses were about 0.05, 0.5, and 5 mg/kg. When measured in MP particle number per kg of animal body weight a day, the doses ranged from $3.5 \times 10^3$ to $1.5 \times 10^{14}$, with the median of $2.5 \times 10^8$ particles/kg/day. According to *Cox et al. (2019)*, an average human consumes 39,000 to 52,000 particles/year with water and food, *i.e.*, about 1.5–2 particles/kg/day. In the described experiments on mice and rats, the concentration of MP in drinking water varied from $7.2 \times 10^3$ to $1.2 \times 10^{18}$ particles/m$^3$, with the median of $1.2 \times 10^{11}$ particles/m$^3$. The "natural" MP concentrations in marine and fresh waters range from 0.1 to $2 \times 10^4$ particles/m$^3$ (*Erni-Cassola et al., 2019*; *Sarijan et al., 2021*). Thus, MP concentrations used in the published experiments involving mice and rats represented, on average, a $10^6$-fold excess compared with the MP content in natural waters and industrial foods.

**Table 1 Experimental research on the effect of microplastic (MP) on the health of laboratory rodents.**

| Publication | Animals | Type and size of MP particles, additional treatment | Dose* | Exposure | The results of the impact of MP |
|---|---|---|---|---|---|
| Replacing water in drinkers with MP suspension | | | | | |
| Luo et al. (2019b) | Pregnant female ICR mice | PS 0.5 and 5 μm | 0.024 and 0.24 mg/kg/day | Throughout the entire pregnancy (3 weeks) | Disorders of fatty acid metabolism were observed in the offspring of mice that consumed MPs. |
| Lu et al. (2018b) | Male ICR mice | PS 0.5 and 50 μm | 0.024 and 0.24 mg/kg/day | 5 weeks | MP causes a decrease in body weight, the relative mass of the liver and adipose tissue; decreased mucin secretion and expression of Muc1 and Klf4 in the colon; significant changes in the composition of the intestinal microflora; disorders of lipid metabolism in the liver. |
| Hou et al. (2021b) | Male ICR mice | PS 5 μm | 0.024, 0.24 and 2.4 mg/kg/day | 5 weeks | Under the influence of MP, there was the decrease in the number of viable spermatozoa in the epididymis, an increase in the proportion of deformed spermatozoa, atrophy and apoptosis of spermatozoa in the testes, an increase in the expression of pro-inflammatory markers: NF-κB, IL-1β, IL-6, a decrease in the expression of the anti-inflammatory molecule Nrf2/HO-1. |
| Jin et al. (2019) | Male ICR mice | PS 5 μm | 0.024 and 0.24 mg/kg/day | 6 weeks | MP accumulates in the intestine, causes a disturbance of the intestinal barrier, changes in the intestinal microflora, disturbances in the metabolism of bile acids. |
| Luo et al. (2019a) | Pregnant female ICR mice | PS 5 μm | 0.024 and 0.24 mg/kg/day | Throughout the pregnancy and lactation (6 weeks) | In the offspring of mice that received MP during gestation and feeding, metabolic disorders in the liver and changes in the composition of the intestinal microflora are detected. |
| Shengchen et al. (2021) | Male C57BL/6 mice | PS 1–10 μm and 50–100 μm | 2.4 mg/kg/day | 8 weeks (On the 25th day, the tibialis anterior muscle was injured by the injection of BaCl2, 30 days after the muscle injury animals were withdrawn from the experiment) | MP consumption led to overproduction of ROS, the development of oxidative stress, and impaired skeletal muscle regeneration. MP suppressed myogenic and stimulated adipogenic differentiation of myosatellite cells. Muscle regeneration was negatively correlated with MP particle size. |

(Continued)

| Publication | Animals | Type and size of MP particles, additional treatment | Dose* | Exposure | The results of the impact of MP |
|---|---|---|---|---|---|
| *Zheng et al. (2021)* | Male C57 mice | PS 5 μm, induction of acute ulcerative colitis with 3% sodium dextran sulfate (DSS) solution in drinkers | 0.12 mg/kg/day | 7 days | MP exacerbates the DSS-induced acute colitis; causes dystrophic changes in the liver. |
| *Deng et al. (2018)* | Male CD-1 mice | PS and PE 0.5–1 μm contaminated with organophosphate fire retardants (OPFR) | 0.48 mg/kg/day | 13 weeks | MP and OPFR together exhibited more pronounced effects than either separately: oxidative stress, neurotoxicity, impaired amino acid metabolism and energy metabolism. |
| *An et al. (2021)* | Female Wistar rats | PS 0.5 μm | 0.06, 0.6 and 6 mg/kg/day | 13 weeks | MP is detected in ovarian granulosa cells, causes their apoptosis and the development of ovarian fibrosis. |
| *Hou et al. (2021a)* | Female Wistar rats | PS 0.5 μm | 0.015; 0.15 and 1.5 mg/kg/day | 13 weeks | MP is detected in ovarian granulosa cells, causes pyroptosis and apoptosis of these cells. |
| *Li et al. (2021)* | Female Wistar rats | PS 0.5 μm | 0.06, 0.6 and 6 mg/kg/day | 13 weeks | MP causes damage to the seminiferous tubules, apoptosis of spermatogenic cells, a decrease in sperm motility, an increase in the proportion of abnormal spermatozoa. |
| *Li et al. (2020b)* | Female Wistar rats | PS 0.5 μm | 0.05, 0.5 and 5 mg/kg/day | 13 weeks | MP causes oxidative stress in the myocardium, apoptosis of cardiomyocytes, cardiosclerosis and cardiac dysfunction. |
| *Wei et al. (2021)* | Female Wistar rats | PS 500 μm | 0.05, 0.5 and 5 mg/kg/day | 13 weeks | MP consumption leads to disruption of the structure and function of the heart. Damage of mitochondria in cardiomyocytes and death of these cells are noted. Levels of creatine kinase-MB and cardiac troponin I (cTnI) are elevated. |
| Administration of MP suspension through a gastric tube | | | | | |
| *Qiao et al. (2021)* | Male C57/B6 mice | PS 0.07 μm (NP) and 5 μm, unmodified, negatively charged carboxylated and positively charged aminated | 0.2 and 2 mg/kg/day | 4 weeks | MP caused intestinal damage, a decrease in the expression of tight contact proteins in the intestinal epithelium, and pronounced changes in the intestinal microflora. |
| *Jin et al. (2021)* | Male Balb/c mice | PS 0.5 μm, 4 μm and 10 μm | 40 mg/kg/day | 4 weeks | MP particles, 4 and 10 μm in diameter, are detected in the testes one day after the first injection. On the 28th day of exposure, a decrease in testosterone levels and sperm quality is observed. Spermatogenic cells die and are arranged randomly, multinucleated gonocytes appear in the seminiferous tubules. |

| Publication | Animals | Type and size of MP particles, additional treatment | Dose* | Exposure | The results of the impact of MP |
|---|---|---|---|---|---|
| Stock et al. (2019) | Male genetically modified C57BL/6 mice | PS 1 μm, 4 μm and 10 μm | 1, 63 and 33 mg/kg/day according to size | 4 weeks | In animals getting MP, body and organ weight did not change, there were no signs of oxidative stress or inflammation in the intestine. |
| Wang et al. (2021) | Male C57BL/6 mice | PS 2 μm | 8 and 16 mg/kg twice a week | 4 or 8 weeks | MP accumulates in the kidneys, causing structural damage. In the kidney, levels of ER stress, the production of inflammatory markers and proteins associated with autophagy are increased. |
| Sun et al. (2021) | Male C57BL/6 mice | PS 5 μm | 4 and 20 mg/kg/day | 4 weeks | MP affects hematopoiesis. Decreases the number of leukocytes and the CFU-GM, CFU-M, CFU-G; changes 41 (lower dose) or 32 (large dose) genes in bone marrow cells. |
| Deng et al. (2017) | Male ICR mice | PS 5 and 20 μm | 4 mg/kg/day | 4 weeks | The maximum concentration of MP in the liver, kidneys, and intestines is reached by the 14th day of the experiment. The relative weight of the liver decreases at a MP dose of 0.5 mg/day. In the liver, inflammatory changes and fatty degeneration are observed. Disorders of energy and lipid metabolism, oxidative stress were revealed. |
| Yang et al. (2019) | Male mice | PS 5 and 20 μm | 0.4, 4 and 20 mg/kg/day | 4 weeks | Toxicokinetic/toxicodynamic study of MP influence. The accumulation of MPs in the liver, kidneys, and intestines was assessed over time. |
| Xie et al. (2020) | Male Balb/c mice | PS 5–5.9 μm | 0.4; 4 and 40 mg/kg/day | 6 weeks | MP causes a decrease in spermatozoa number and mobility, an increase in the proportion of deformed spermatozoa; a decrease in the activity of the enzymes succinate dehydrogenase and lactate dehydrogenase; decrease in testosterone content, development of oxidative stress. |
| Rafiee et al. (2018) | Male Wistar rats | PS 0.025 and 0.05 μm (NP) | 1, 3, 6 or 10 mg/kg/day | 5 weeks | In neurobehavioral tests, statistically significant changes were not observed upon exposure to MP, body weight did not change. |

(Continued)

| Publication | Animals | Type and size of MP particles, additional treatment | Dose* | Exposure | The results of the impact of MP |
|---|---|---|---|---|---|
| *da Costa Araújo & Malafaia (2021)* | Male Swiss mice | PE 35.46 ± 18.17 μm | 4.8 mg/kg/day | 1 week | In animals that consumed MP, a decrease in locomotor activity and a higher anxiety index in the open field test, a lack of protective social aggregation, and behavior with a reduced risk assessment when meeting a potential predator were observed. |
| *Park et al. (2020)* | Male and female ICR mice | PE 40–48 μm modified with acid and hydroxy groups | 3.75, 15 and 60 mg/kg/day | 13 weeks | MP caused reactions from the immune system in adult animals: in mice of both sexes, the content of neutrophils in blood increased, in females, the content of IgA in blood increased, and the subpopulation composition of lymphocytes in the spleen changed. In animals receiving MP, the number of live births per female and the body weight of newborn pups decreased significantly. |
| *Deng et al. (2021)* | Male CD-1 mice | PE 0.4–5 μm, phthalate-contaminated | 100 mg/kg/day | 4 weeks | MP can penetrate the testes of mice. MPs with phthalates accumulate in the liver, intestines, and testes. MP enhances the reproductive toxicity of phthalates. |
| *Deng et al. (2020)* | Male CD-1 mice | PE 45–53 μm, phthalate-contaminated | 100 mg/kg/day | 4 weeks | MP can transport and release phthalates into the intestines of mice. MP enhances the toxic effects of phthalates: increased intestinal permeability, oxidative stress, inflammatory reactions, metabolic disorders. |
| **MP in food** | | | | | |
| *Li et al. (2020a)* | Male C57BL/6 mice | PE 10–150 μm | 0.24, 2.4 and 24 mg/kg/day | 5 weeks | MP caused changes in the composition and diversity of intestinal microflora, an increase in the level of IL-1α in the blood serum, an increase in the proportion of Th17 and Treg cells among CD4+ cells. MP in a high dose caused the development of inflammation in the small intestine. |
| **Intratracheal introduction of MP** | | | | | |
| *Fournier et al. (2020)* | Pregnant female Sprague Dawley rats | PS 0.02 μm (NP) | $2.64 \times 10^{14}$ MP particles | 1 time on day 19 of gestation, removal from the experiment in a day | MP particles were detected in maternal lungs, heart and spleen. MP was detected in the placenta, as well as in the liver, lungs, heart, kidneys and brain of fetuses, which indicates translocation of MPs from the mother's lungs to the fetal tissue in late pregnancy. |

(Continued)

| Table 1 (continued) | | | | | |
|---|---|---|---|---|---|
| Publication | Animals | Type and size of MP particles, additional treatment | Dose* | Exposure | The results of the impact of MP |
| MP inhalation | | | | | |
| Lim et al. (2021) | Male and female Sprague-Dawley rats | PS 0.1 μm (NP) | MP air concentration $0.75 \times 10^5$, $1.5 \times 10^5$ and $3 \times 10^5$ particles/sm$^3$ | 2 weeks | Under the influence of MP, the increase in the relative mass of the heart, a decrease in the content of leukocytes and lymphocytes in the blood, a decrease in the time of inspiration were revealed, furthermore a tendency to an increase in the content of cytokines TGF-β and TNF-α in the lung tissue was observed. |
| Intraperitoneal injection of MP | | | | | |
| Estrela et al. (2021) | Male Swiss mice | PS 0.023 μm (NP) | 14.6 ng/kg | 3 days | MP causes cognitive impairments, violations of the redox balance, and a decrease in the activity of acetylcholinesterase in the brain. |

**Notes:**
* In articles, when MP is added to drinkers, as a rule, the concentration of MP in drinkers is indicated as mg/l, when administered through a gastric tube - the amount of MP (mg) per animal. We recalculated the MP dose as mg of MP per kg of animal's weight per day, taking the average weight of a mouse equal to 25 g, a rat - 250 g, daily water consumption in a mouse - 6 ml, in a rat - 25 ml.
NP – nanoplastic.

The duration of exposure varied from a single injection of MP to a continuous exposure over three months; most typically, the exposure lasted four weeks. The high variability of experimental parameters questions the possibility of reliable comparison between the studies.

When administered orally, MP primarily act on the gastrointestinal tract. A number of studies demonstrate accumulation of 0.5–50 μm PS granules in the colon of mice, causing damage to the colon epithelial barrier, reducing the mucus production, inhibiting the expression of tight-junction proteins, and altering the intestinal microflora composition (Lu et al., 2018b; Jin et al., 2019; Li et al., 2020a; Qiao et al., 2021). MP were also detected in the rodents' liver and kidneys, causing inflammatory changes and oxidative stress, a decrease in the relative mass of the liver, and energy and lipid metabolism imbalances (Deng et al., 2017; Deng et al., 2018; Yang et al., 2019; Deng et al., 2021; Wang et al., 2021). MP have been implicated in the development of cardiac fibrosis and impaired cardiac function by promoting cell death of cardiomyocytes through the induction of oxidative stress and mitochondrial damage (Li et al., 2020b; Wei et al., 2021). The mice administered with MP revealed abnormal patterns of skeletal muscle regeneration, with suppressed myogenic differentiation of myosatellite cells and their redirection towards adipogenic differentiation (Shengchen et al., 2021). MP also affect hematopoiesis by suppressing the leukocyte differentiation and altering expression of dozens of genes in the bone marrow cells (Sun et al., 2021). MP exposure has been shown to promote cognitive impairments and affect animal behavior (da Costa Araújo & Malafaia, 2021; Estrela et al., 2021), and also to cause reproductive consequences and developmental disorders in the offspring. In male mice, MP penetrated into the testes, caused damage to

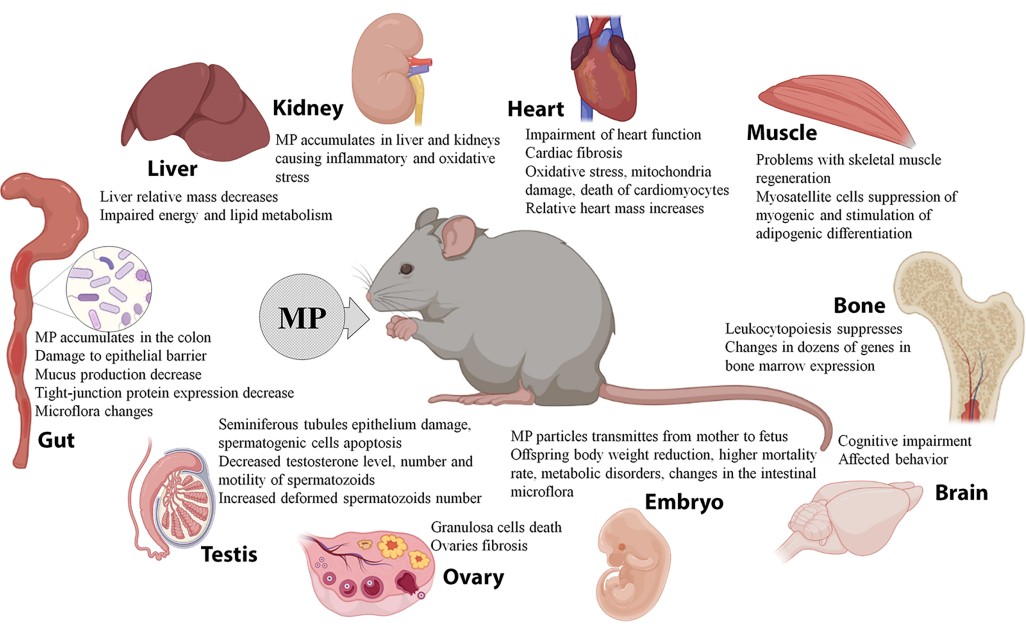

**Figure 2 Effects of microplastics (MP) on mice and rats.**

the epithelium of seminiferous tubules and apoptosis of spermatogenic cells, while reducing testosterone levels along with the number and motility of spermatozoa and increasing the number of dysfunctional deformed spermatozoa (*Xie et al., 2020*; *Hou et al., 2021a*; *Jin et al., 2021*; *Li et al., 2021*). In female mice, MP penetrated into the ovaries, causing granulosa cell death and fibrosis (*An et al., 2021*; *Hou et al., 2021b*). The offspring of mice that consumed plastics had reduced body weight, higher neonatal mortality rates, metabolic imbalances, and altered intestinal microflora composition (*Luo et al., 2019a*; *Luo et al., 2019b*; *Park et al., 2020*). MP also enhanced the toxic and pathological effects of organophosphates, antiperenes, phthalates, and sodium dextran sulfate (*Deng et al., 2018*; *Deng et al., 2020*; *Deng et al., 2021*; *Zheng et al., 2021*).

Two studies evaluated the effects of MP that entered the body through the airways by inhalation or intratracheal administration. The airborne MP significantly increased the relative mass of the heart while reducing the leukocyte and lymphocyte counts and shortening the inspiration time. The treatment also led to increased production of TGF-β and TNF-α cytokines in the lung. Moreover, the particles were effectively transmitted to the fetus: MP were detected in the placenta, as well as embryonic livers, lungs, hearts, kidneys, and brains (*Fournier et al., 2020*; *Lim et al., 2021*).

At the same time, several studies demonstrated the lack of significant effects of the MP exposure. According to *Stock et al. (2019)*, mice treated with MP had unaltered body and organ weights and no signs of inflammation or oxidative stress in the intestine. *Rafiee et al. (2018)* observed the same results of neurobehavioral tests in the rats that consumed MP and control animals.

Overall, the experiments on laboratory rodents indicate effective spreading of MP, which enter the body with water, food, and/or air, to various organs and tissues. The MP

particles were detected in the intestines, livers, kidneys, lungs, spleens, hearts, ovaries, and testes of the animals, causing biochemical changes, structural damage, and dysfunction. MP can effectively cross the placental barrier and interfere with offspring development. In addition, MP may absorb and carry various pollutants and enhance their negative effects (Fig. 2). However, due to the small number of available studies and significant variation in particle doses, sizes, and the exposure parameters, the data are still fragmentary and controversial. It should be noted that the doses of MP conventionally used in animal studies significantly exceed the content of MP in natural samples and consumer products.

## CONCLUSIONS

Every year, we generate millions of tons of plastic waste. Hardly a quarter of this amount is recycled and/or disposed of properly. The accumulating plastic waste pulverizes into small particles polluting the environment and spread by wind and water. Due to the ubiquitous pollution of habitats, water, food, and inhalable air, the microplastics pervade and accumulate in the bodies of living organisms including humans. Experimental studies on invertebrates, fish, mice, and rats reveal the negative effects of microplastics on health and implicate them in pathological changes of different organs. According to recent findings, environmental pollution with microplastics poses a significant threat to human health. Still, the evidence regarding the scale of this threat is ambiguous and fragmented. The impact of particular parameters of microplastics (*e.g.*, the sorption capacity for pathogens, stage of decomposition, *etc.*) remains uncertain. Clarification of these issues will require further research efforts.

### Funding
The work was supported by the Russian Science Foundation (No. 22-24-00232). The funders had no role in study design, data collection and analysis, decision to publish, or preparation of the manuscript.

### Grant Disclosures
The following grant information was disclosed by the authors:
Russian Science Foundation: 22-24-00232.

### Competing Interests
The authors declare that they have no competing interests.

### Author Contributions
- Natalia Zolotova conceived and designed the experiments, analyzed the data, prepared figures and/or tables, and approved the final draft.
- Anna Kosyreva performed the experiments, analyzed the data, authored or reviewed drafts of the article, and approved the final draft.

- Dzhuliia Dzhalilova performed the experiments, prepared figures and/or tables, and approved the final draft.
- Nikolai Fokichev performed the experiments, prepared figures and/or tables, and approved the final draft.
- Olga Makarova conceived and designed the experiments, analyzed the data, authored or reviewed drafts of the article, and approved the final draft.

### Data Availability

This article is a literature review.

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
