# Peer review of "Harmful effects of the microplastic pollution on animal health: a literature review"

_PeerJ, doi:10.7717/peerj.13503_

## Round 0.1 · original submission · Major Revisions

Three reviewers have provided their opinions and suggestions for improving the manuscript. Please consider the suggestions and make necessary changes in the manuscript. All reviewers have pointed out the need to better define the scope of the review, by rephrasing the title and editing the introduction. Please pay special attention to this important aspect when revising the manuscript.

·

Basic reporting

The english language and writing quality is unequal along the paper. The introduction especially needs particular attention:
L51 "The" term plastic
L57 1mm thick
L82 "different classes of animals" or "differentanimals"
L93: terms «soil», «air», «water», or «ocean» were used in the query
L114 microparticles of any polymer type
L140 I suggest “Microplastics Environmental pollution”
L323 define ICR at first occurrence

Literature is sometimes missing or not cited early enough.
L165 References for the content of plastics in River?
L176 reference of the article in sweden?
L180 reference for the MP in the air?
L211 refernce for the MP amount per year
L213, is this a percentage you calculated among all studies ? Or did you fin in a review?
L239 Give the name of the review in the first sentence of the paragraph

The structure is globally fine. Some minor change could be made:
L72. I recommend you first introduce the plastic composition with main polymers and additives and then the properties.
L.201 I would present the MP in food and finish with the human feces and placenta

The review is of broad and cross-disciplinary interest and within the scope of the journal

The field been reviewed recently. I suggest the authors focus on the plastic content in animals and the potential effects

Additional figures and discussion would increase the interest for the review
L260, add a table and/or figure to describe the effects in fish

L331 discuss the applied doses and related effects in comparison with the environmental content in the matrices (soil, water) and animals you describe previously

L337 It would be nice to have a figure to summarize the effects on mice

Please also discuss other factors leading to toxicity that are not yet tested, for example the transport of sorbed contaminants or pathogens on the plastic particles and the degradation stage of the plastic.

The title should be much more specific as the subject developed by the authors is clearly the effect of microplastics in animals.

Experimental design

The Methods are not described with sufficient detail & information.
L96 How many original papers and reviews were published on these topics?
L118 Why studies until 2017 are selected if you say you analysed from January 1st, 2010 onwards?

Validity of the findings

L65. Why would you include “particles formed as the result of automobile tires wear, some types of road surface” as primary MP. I doubt small plastic particles are deliberately created in these case.


L72. I suggest you also mention physical properties such as elasticity and shear strength. These properties will depend on the plastic polymers so I recommend you first introduce the plastic composition with main polymers and additives and then the properties.

L133 for water, dissolved sea salt and honey I would be surprise a digestion step would have been included. Pease check and give references.

L137 I don’t think you mean “Then” may be “alternatively”: spectral method (FTIR or Raman) and visual counting give both the number of MP

L182 I would not consider cellulose fibers as MP

Additional comments

I thank the authors for their review of this very interesting topic. The authors describe the potential for microplastic in animals and expected harmful consequences. I suggest to define better the scope of the review, in the title and introduction.

Reviewer 2 ·

Basic reporting

In general, the manuscript reviewed few aspects: methods for determining the content of microplastics, microplastics in water, soil and air, microplastics in food chains and human food, hazardous effects of microplastics on the health of different classes animals (Invertebrates, Fishes, Mammals). It has a broad interest, but it didn’t make a solid foundation to estimate the potential risk of microplastics to human health.

The title is too general. As the authors stated in the manuscript, “The aim of this review is to systematize and summarize the results of experimental studies of the microplastics effect on the health of different classes animals and to estimate the potential risk of microplastics to human health.” Why don’t you focus on your aim and only review the hazardous effects of microplastics on the health of different classes animals?

Experimental design

no comment

Validity of the findings

The argument for the goals set out in the Introduction is not well developed. More statements are needed for how to link the results from animals to human.

Additional comments

Line 24, undoubt to undoubted. Line 60, hundreds to hundreds of. Line 95, full stop dot (.) is missing in the end of the sentence.
Please check the number in Line 153 and 157, are they 268.94 and 79.000 or 26894 and 79000?
Lines 185-193, please add references where the data from.
Line 231, please rephrase the subtitle “Microplastic hazard effects on animals”. It’s not correct in gramma.
Line 243, please rephrase “in the presented in the review works”.

·

Basic reporting

The article is conformed to professional standards of courtesy and expression. It is written in clear, unambiguous English. Nonetheless, at line 114 “tested microparticles of any plastics” could be written in a clearer way as well as “Microparticles of plastic” in line 198, that could be simply referred as “Microplastics” or “MP”.

The article includes sufficient field background with appropriate references providing justification for this literature review. As a minor review, please check reference 62 (line 605): a full stop after “2021” is missing.

Improving the description of nanoplastics may be beyond the purpose of this manuscript, nonetheless I suggest adding NP abbreviation at line 61 when “nanoplastics” are first mention so that it is possible to recall them explicitly through the text when it could be used (lines 263, 264, 266, 328 and in Table 1).

The structure of the article is conformed to an acceptable format of ‘standard sections’ of the Journal. However, the title could be more specific, stating that the review is focused on MP effects on animal health with special attention on mammals or laboratory rodents.

The abstract fulfils its scope, as a minor review, please specify that the “Small particles of plastic” (line 17), are of different types.

The Table 1 is relevant to the content of the article summarising the results from different studies, however it should be improved to make it easier to consult and more exhaustive:

- All the “Ref #” numbers have to be changed to be properly associated with the actual references.
- The “Animals“ column should be compiled in a consistent way through the Table. The colour code “Red text – female animals” needs to be checked in Luo T et al.,2019(a) and (b), Rafiee M et al.,2018, Park EJ et al., 2020 and Lim D et al., 2021. “Male” has to be added in Lu L et al., 2018 and Jin Y et al., 2019.
- The heading of the column “Type and size of MP particles” should be stated in a more suitable way since in some cases different treatment and NP are included. When NP are used it should be made explicit (Qiao J et al., 2021, Rafiee M et al., 2018, Fournier SB et al., 2020 and Estrela FN et al., 2021).
- Please use English alphabet in the “Dose” column in Xie X et al., 2020.

The review is of broad and cross-disciplinary interest, as it is also stated by the authors in lines 27-28. Although the field has been reviewed recently enough, the value of this article, that describes the field and explains why it is needed, stays in the different point of view that provides. As a matter of facts, it considers the results from the most recent papers about the study of MP effects in vivo, on laboratory rodents, which can give new insights on effects on human health as it is made clear since the introduction (lines 83-87).

Experimental design

The Survey methodology is well described. It is reproducible, the used criteria ensure comprehensive and unbiased coverage of the field and the assessment of the updated state of the art. The sources are appropriately cited and the manuscript is logically organized into subsections. As minor revisions:

- Details about the MP entrance mechanism in the women placenta (Ragusa et al., 2021) (lines 203-204) can be added.
- In lines 321-322 can be mentioned that different types of MP and NP are also taken in account in Table 1.
- In line 328 can be indicated that plastics particle with a diameter of 0.02 are considered NP.

Validity of the findings

The conclusions are appropriately stated and connected to the original question investigated. Nonetheless they can be expanded. It can be discussed if the doses tested in the in vivo studies are comparable to the MP potential exposure in the natural environment and/or giving an indication for approaches and future directions for further research.

---

## Round 0.2 · Minor Revisions

Thank you for revising the manuscript. The Reviewers were mostly satisfied with the changes made, however, they identified some language issues. Please correct the language and resubmit.

·

Basic reporting

Despite some improvements, some part of the text are still difficult to read because of the grammar. I listed some errors below but please ask a fluent speaker to carefully review all the document.

L16 “Small particles of different plastic types”
L17. Connect the two sentences : “Small particles of different types plastic with a diameter of less than 5 mm are found around the globe in water, air, soil, and various living organisms. An individual constantly absorbs microplastics with water and food, inhales with air.

L21. I think this sentences are un-necessary in the abstract: “Experimental studies of the potential health risks of microplastic are now under the undoubted scientific interest. The number of publications on this topic is growing every year. However, it could be difficult to analyze the potential data because of narrow investigations and contradictory results obtained, which should be combined and generalized.

l26 "of different classes animals"

Figure 1. Brain “behavior in shallow water” is not connected to the rest of the sentence.

Figure 2 connect the sentence “MP accumulates in liver and kidneys causing inflammatory and oxidative stress” or remove verbs “MP accumulation in liver and kidneys, inflammatory and oxidative stress” and harmonized in all the figure.

Figure 2 I suggest to highlight the specific organs in the text with bold, as it is done in the Figure 1

l. 403. “Plastic waste broken into small particles of MP, which carried by water and wind” verbs are missing
l404: “particles of microplastics with food” reformulate. Suggestion “Water, food, and inhaled air are pathways for microplastics to penetrate and accumulate in the bodies of various living organisms, including human.”

L407: “can provide” -> “can have”

Experimental design

The citation has been improved.

Validity of the findings

Table 1 and Figure 1 and 2 are good tools to systematize and summarize the microplastics effect on the health

·

Basic reporting

The article is conformed to professional standards of courtesy and expression. It is written in a clear, unambiguous English.
Could you please ensure that “MP” is written in a consistent way through the text? A couple of times it is written “MPs” and it is not clear to me if it is done just by mistake or as a purpose.

The structure of the article is conformed to an acceptable format of ‘standard sections’ of the Journal. The title is specific, stating that the review is focused on MP effects on animal health.
The abstract fulfils its scope.
The Table is relevant to the content of the article summarising the results from different studies.
The Figures describe appropriately the MP effects on animals.

Experimental design

no comment

Validity of the findings

no comment

---

## Round 0.3 · accepted · Accept

Thank you for revising the manuscript. It is now suitable for publication.